

# Downscaling land use and land cover from the Global Change Assessment Model for coupling with Earth system models.

Yannick Le Page[1,2], Tris O'Brien West[1], Robert Link[1], Pralit Patel[1]

[1] Centro de Estudos Florestais, Instituto Superior de Agronomia, Universidade de Lisboa Department Tapada da Ajuda, 1349-017 Lisbon, Portugal
[2] Pacific Northwest National Laboratory, Joint Global Change Research Institute, University of Maryland, College Park, MD 20740, USA

*Correspondence to*: Yannick Le Page (niquya@gmail.com)

**Abstract.** The Global Change Assessment Model (GCAM) is a global integrated assessment model used to project future societal and environmental scenarios, based on economic modeling and on a detailed representation of food and energy
production systems. The terrestrial module in GCAM represents agricultural activities and ecosystems dynamics at the sub-regional scale, and must be downscaled to be used for impact assessments in gridded models (e.g. climate models). In this study, we present the downscaling algorithm of the GCAM model, which generates gridded time series of global land use and land cover (LULC) from any GCAM scenario. The downscaling is based on a number of user-defined rules and drivers, including transition priorities (e.g. crop expansion preferentially into grasslands rather than forests) and spatial constraints
(e.g. nutrient availability). The default parameterization is evaluated using historical LULC change data, and a sensitivity experiment provides insights on the most critical parameters and how their influence changes regionally and in time. Finally, a reference scenario and a climate mitigation scenario are downscaled to illustrate the gridded land use outcomes of different policies on agricultural expansion and forest management. Several features of the downscaling can be modified by providing new input data or changing the parameterization, without any edits to the code. Those features include spatial resolution as
well as the number and type of land classes being downscaled, thereby providing flexibility to adapt GCAM LULC scenarios to the requirements of a wide range of models and applications. The downscaling system is version controlled and freely available.

## 1 Introduction

The Global Change Assessment Model (GCAM) has been developed to better understand interactions between natural and
human systems and anticipate their co-evolution in the future. It combines representations of the global economy, energy systems, agriculture and land use with a representation of terrestrial, ocean and atmospheric biogeochemical, ice-melt and



climate processes (Clarke et al., 2007; Kim et al., 2006). GCAM is extensively used to explore the direct effects of changes in exogenous assumptions, such as population, technology, yield improvements, economic and environmental policies, as well as their system-wide repercussions. These analyses are performed using GCAM as a standalone model or coupled to specialized models with greater capabilities for impact assessments. For example, GCAM provided greenhouse gas

emissions and land use/land cover (LULC) projections for one of the 4 scenarios of the IPCC 5[th] Assessment Report, the Representative Concentration Pathways 4.5 (RCP4.5, (Thomson et al., 2011, p.4)). Those data were used as inputs to climate models to generate climate projections and perform a wide range of impact, adaptation and vulnerability assessments (Pachauri et al., 2014).

Coupling models involves adapting the data and models so that the flow of information is consistent with the source model and can be assimilated by the destination model. In the case of GCAM, spatial resolution is a technical challenge for coupling studies looking at the impacts of LULC change. GCAM represents these dynamics at sub-regional scale while Earth system models and regional natural resource (i.e., agriculture and forestry) models typically operate at gridded scales. For example, the U.S. is divided into 8 agro-ecological zones (AEZs, (Monfreda et al., 2009)), of various sizes (up to 2.5

Million km$^2$) and shapes. GCAM tracks the share of land categories in each AEZ, but not where they are actually located within each zone. This issue was addressed for the IPCC 5[th] Assessment Report with a downscaling algorithm designed specifically for coupling to climate models (Hurtt et al., 2011). However, gridded LULC scenarios are needed for a variety of models and applications, each requiring specific data attributes (e.g. resolution, land categories, management practices (Kraucunas et al., 2014); Huang et al., in prep.). A flexible downscaling component within the GCAM modeling framework

is thus needed to meet these multiple applications.

This paper presents LULC downscaling capabilities, code, and parameters developed for the GCAM model. An evaluation analysis with historical land use data is performed to quantify the spatial accuracy of land use change allocation, and a sensitivity analysis is conducted to identify critical parameters. Finally, the downscaling method is applied to scenarios of

future land use change projected by GCAM, generating gridded land use projections that can be used for impact assessments in coupling experiments.

## 1 Methods

### 2.1 Overview of the terrestrial module in GCAM

The terrestrial system is represented in GCAM to account for its role in food, wood, and energy production (Calvin et al.,

2014) and in the carbon and water cycles (Hejazi et al., 2014; Le Page et al., 2013), providing capabilities to explore interactions and the implications of environmental policies (Figure S1 in Supp. Material; (Kraucunas et al., 2014; Thomson



et al., 2011). While we focus here on describing the spatial scale and land types in GCAM because they are essential aspects for the downscaling, a more technical description of the terrestrial module is available in the literature (Kyle et al., 2011; Wise et al., 2014; Wise and Calvin, 2011).

GCAM represents the world terrestrial biosphere into 283 spatial units, the result of the intersection of two spatial scales. First, the world countries are aggregated into 32 geo-political and socio-economic regions (Figure 1a), a scale at which most economic sectors are represented (e.g. industrial production, energy use, trade, and natural resources). Second, global land area is split into 18 agro-ecological zones (AEZs, Figure 1b) to represent natural ecosystems and agricultural activities, providing a climate-based zoning to better account for vegetation and crop productivity. These 18 AEZs are intersected with
the 32 regions to get the 283 unique combinations of region and AEZ.

Each of these 283 region/AEZ can have up to 22 types of land cover: five types of natural ecosystems and 17 types of managed lands (Table 1). Over the spin-up period (1700-2005), those land areas are inferred from the History Database of the Global Environment (HYDE, Klein Goldewijk et al., 2011), the FAO RESOURCESTAT database (FAO, 2010) and
potential vegetation data (Ramankutty and Foley, 1999a, as detailed in (Kyle et al., 2011). For future time steps, GCAM integrates a range of drivers to determine LULC change, including food demand (population growth, diet changes), region/AEZ level crop productivity and costs (e.g. labour, fertilizer), energy demand (biomass crops) and environmental policies, among others. A major application of GCAM consists in exploring LULC projections under alternative configurations of these drivers (Calvin et al., 2014; Thomson et al., 2010, 2011). For example, while the reference scenario
projects continuing deforestation and expansion of global agriculture in response to growing food demand, implementing terrestrial carbon market shifts the economics of land use decisions towards agriculture intensification and afforestation. These scenarios are depicted at the region/AEZ scale, but can be gridded with the downscaling method detailed below.

## 2.2 Downscaling method

### 2.2.1 Overview

The LULC downscaling method is based on previous work by (West et al., 2014) at the national scale for the U.S. It consists of allocating tabular land areas from a relatively coarse set of spatial units to a higher resolution land cover grid (West et al., 2010). In the case of GCAM, these spatial units are defined by the intersection between regions and AEZs (Figure 1), and the resolution of the final grid for the global analysis presented here is 0.25 degrees (~28km at the equator). The algorithm relies on gridded observations of land use and land cover and on a set of user-defined rules to spatially allocate land types within
each region/AEZ (Figure 2).



The algorithm starts with pre-processing of the GCAM and gridded LULC data to harmonize their attributes. It is labeled the reconciliation phase, and consists of (1) aligning total land areas of each region/AEZ from both data sources, and (2) converting their respective land schemes to a common set of land types that are ultimately downscaled.

Following data reconciliation, the actual downscaling is performed. The algorithm starts in the base-year (first time step, e.g. 2005), and modifies the observed gridded data until land shares in each region/AEZ are equal to the shares in the GCAM model. If a given region/AEZ has 100km$^2$ of crops in GCAM but only 80km$^2$ in the observation, 20km$^2$ of crops will be created by converting other land types that are in excess. The geospatial allocation of the additional 20km$^2$ of cropland depends on a number of downscaling rules that are described in the next section, including land use transition priorities and

spatial constraints (e.g., nutrient availability for agriculture). Once completed, the base-year downscaled data consists in gridded LULC that is consistent with GCAM at the region/AEZ scale with spatial patterns similar to observed LULC (Figure 2). This gridded GCAM LULC in the base-year then becomes the starting point to which LULC change of the following time step is attributed, following the same downscaling rules. This process is repeated for each time step until the entire scenario is downscaled (e.g. 2005-2100).

The downscaling system is available on the GitHub open-source software site[1]. The repository includes the source code (written in Python 2.7) and a detailed user manual, including software/library requirements. A reference LULC scenario is provided as part of the downscaling system, as well as spatial LULC data from MODIS for the 2005 base-year. Alternative scenarios can be generated with GCAM, which is available from the Joint Global Change Research Institute[2].

The next sections describe the code and its user-defined parameters in more details. For the sake of clarity, it is based on a relatively simple configuration of the downscaling, referred to as "basic configuration", but many aspects are flexible as specified in the text. The spatial LULC data used in the basic configuration are from the MODIS MCD12Q1 version 5.1 product for the year 2005, PFT Type 5 classification (Friedl et al., 2010), aggregated from 500m to 0.25 degree resolution.

### 2.2.2 Reconciliation phase

2.2.2.1 Matching land areas

Although the downscaling aims at maintaining consistency with the original GCAM land outputs, the total land area in each region/AEZ has to be adjusted to match the observation data. Otherwise, expansion on water (case of more land in GCAM) and removal of observed terrestrial land (case of less land) would be necessary, which was considered unrealistic. All GCAM land types (GLTs) are adjusted by the same ratio within a given region/AEZ:

---

[1] https://github.com/JGCRI/GCAMLU

[2] http://www.globalchange.umd.edu/models/gcam/download/



$$A_{R,Z,L}^{GC} = A_{R,Z,L}^{G} \times \frac{A_{R,Z}^{O}}{A_{R,Z}^{G}} \tag{1}$$

Where $A^G$ and $A^O$ are the GCAM and observed areas, $A^{GC}$ the final, adjusted GCAM areas, $R$ and $Z$ the region/AEZ and $L$ the GCAM land type considered. When compared to MODIS, total area of most regions/AEZs differs by less than 3%. Globally,

5 total land area is 126.9 Million km2 in GCAM and 128.1 Million km2 in MODIS (not including small islands and other territories that are not represented in GCAM and thus remain equal to observations throughout the downscaling).

2.2.2.1 Aggregation to common land type categories

Once total land areas are the same, both the GCAM and spatial land types (GLTs and SLTs) are aggregated to a common

10 scheme of final land types (FLTs). For example, the MODIS data have only one cropland PFT, while GCAM has 13 different crop types. All GCAM crop types are thus aggregated to a single crop category to enable the downscaling: if GCAM corn was kept as an individual land type, there would be no indication of where those corn crops should be allocated from the MODIS data. In the basic configuration, the common scheme includes 7 FLTs: forests, shrubs, grass, crops, urban land, snow, and sparse vegetation.

Aggregation of the spatial data is user-defined with an input table (Table 2). For a given SLT, a number from 0 to 1 determines the share of that land type that goes to each of the FLTs. Typically, the share is either 0 or 1, meaning that an SLT is entirely attributed to a single FLT (Table 2). However, some land cover classifications products include mixed land types (e.g. mosaic of crops and forests). In such a case, the area under that type could be split into crops and forests by using

20 shares of 0.5.

Aggregation of the GCAM data at the region/AEZ scale follows the same concept (Table 3), only with a different approach when a given GLT has to be split into 2 or more FLTs: the shares received by each FLT are determined by the spatial data in that region/AEZ. For example, the GLT *RockIceDesert* can qualify to both the *snow* and the *sparse* FLTs. However, the

25 fraction that should go to each depends on the region/AEZ considered: most would go to *snow* in Greenland, and to *sparse* in central Australia. The table is thus filled with 1s for both (Table 3) to indicate that this GLT needs to be split, and the code computes the actual split as the share of *snow* and *sparse* seen in the spatial data (e.g. MODIS) for each region/AEZ.

**2.2.3 Downscaling rules**

Once the GCAM and spatial data are reconciled and aggregated to the same land type categories (i.e. FLTs), downscaling is

30 performed based on a set of user-defined rules: a "treatment order" defining which FLTs are downscaled first, an "intensification versus expansion ratio", "transition priorities" defining what type of land swaps are favored, and "spatial



constraints" which attribute to each grid-cell a likelihood to receive an expanding FLT. All these rules influence the spatial patterns of the final downscaled GCAM data (see sensitivity analysis in Sect. 1.3).

2.2.3.1 Treatment order

The algorithm downscales all FLTs one after the other; an order thus has to be defined. The input files include a table where users specify that order (Table 4).

2.2.3.2 Intensification ratio

When the GCAM projections indicate that the area of a given FLT is increasing, the additional area can be downscaled on

grid-cells where the FLT already exists – which is referred to as *intensification*, or on grid-cells where it does not yet exist – referred to as *expansion*. In the real world, the ratio of intensification versus expansion varies in space and time. In North America for example, land giveaways, infrastructure development and a number of other factors led to a large-scale westward *expansion* of agricultural activities from 1800 to 1950, then to their *intensification* until today, with most of the Corn Belt now featuring more than 80% crop cover (Ramankutty and Foley, 1999b). In the basic configuration presented

here, the intensification ratio (*intens_ratio*) is set to 0.8, and is part of the sensitivity analysis (Sect. 1.3). The code can easily be modified to define specific ratios for different regions or time periods. Note that the ratio is a target, which sometimes cannot be met. In the extreme case where croplands exist in all grid-cells of a region/AEZ for example, *expansion* is impossible. The code then applies the desired *expansion* target as *intensification* instead.

2.2.3.3 Transition priorities

At each time step, GCAM computes LULC change at the region/AEZ scale, but does not give any indication on land use transitions. For example, if crops and forests increase while shrubs and grass decrease, the share of each possible conversion - or transition - is not known (shrubs to crops, grass to crops, shrubs to forests, grass to forests). It is however an aspect we have to represent when downscaling LULC change to a spatial grid, and is also relevant information for Earth System

modelers (see Discussion). A preference order for land use transitions is thus user-defined in the parameter files, for each FLT (Table 5). In the basic configuration, crops are set to preferentially replace urban land, then grasslands, then shrublands, then forests, etc. This is a preference only: specific transitions can only happen if the FLT to be converted is projected to decrease in GCAM. In the example given above, crops could not be increased into forested land because forests are also projected to increase. This is related to the concepts of "net" versus "gross" LULC change (see discussion).

2.2.3.4 Spatial constraints

Any kind of spatial constraints can be used to influence the downscaling. For a constraint to be implemented, users must provide the input data at the resolution considered and parameterize its influence. The input data must be bound from 0 (fully





constraining) to 1 (no constraint). Then the parameterization defines the relative contribution of that specific constraint in influencing the downscaling, and is specific to each FLT. In the basic configuration illustrated in Table 6, three spatial constraints are listed: kernel density*, soil workability and nutrient availability.

Kernel density represents the proximity and density of a given FLT around a given grid-cell. It is computed by default in the

code, adjusting to the new FLT distribution at each time step. It was implemented under the assumption that new areas of a given FLT tend to appear close to where it already is (e.g. desertification and crop expansion around agricultural areas).

$$KD_{FLT} = \frac{1}{n} \times \sum_{gc=1}^{n} \frac{F_{FLT,gc}}{(D_{gc})^2} \qquad (2)$$

where $KD$ is the kernel density, $n$ the number of neighboring grid-cells included in the computation, $F_{FLT}$ the fraction of the

FLT in the grid-cell considered, and $D$ the distance of that grid-cell to the grid-cell for which the kernel density is being computed. A user-defined parameter – *radius* – represents the size of the moving window used to compute $KD$ for each grid-cell, thus controlling the number $n$. Note that the kernel density depends on land types in the surrounding grid-cells, thus has a different influence on the downscaling than intensification, which is activated based on the considered grid-cell only.

Soil workability and nutrient availability are two indicators of agricultural suitability from the Harmonized World Soil

Database (HWSD, (Fischer et al., 2008)). In Table 6, kernel density contributes 100% of the spatial constrain for all FLTs but for crops, for which it contributes 40% while nutrient availability contributes 40% and soil workability the remaining 20%. Note that the parameterization can include negative numbers to indicate that a constraint has the opposite influence: one could for example favor desertification in low nutrient areas by having a negative value in the cell at the intersection of the "nutrient availability" row and *"sparse"* column in Table 6

### 2.2.4 Code structure and implementation of the downscaling rules to allocate land use change

A stylized overview of the code structure is shown in Figure 3. Not included in this overview are a number of functions which can be activated in the user-input files to perform optional tasks such as mapping FLT distributions, saving statistics and converting the output data (.csv files by default) to netcdf format. A break down of run time and output sizes as a

function of resolution, number of FLTs and optional tasks activated are provided in the user-manual. Downscaling all 5-year time steps of a 2005-2100 scenario at 0.25 degree resolution with 7 FLTs takes a few minutes on a standard desktop computer and produces around 100Mb of output data when all optional tasks are turned off.

To better illustrate how the downscaling is actually computed, we provide a specific example: in a given region/AEZ, crops

are projected by GCAM to increase by 100km$^2$ from 2015 to 2020, while grasslands and forests decrease by 70km$^2$ and





30km$^2$, respectively. All other FLTs remain unchanged. Of the 100km$^2$ cropland increase, 80km$^2$ are set to occur as intensification and 20km$^2$ as expansion (user-defined intensification ratio of 0.8, see Sect. 1.2.3.2).

The code goes through the first intensification loop (Figure 3), following the treatment order (Table 4). Urban, snow and sparse land types are skipped as they remain unchanged, then the loop goes to crops, which have to be intensified by 80km$^2$. The code thus goes through the land use transition priorities for crops (Table 5), starting with urban land, which does not have any area to spare and therefore remains unchanged. The second priority is grass, which does have land to spare as it is projected to decrease by 70km$^2$. All grid-cells in that region/AEZ containing both crops and grass (intensification is for grid-cells which already have crops) are thus selected. They are then attributed an index $S$ from 0 to 1 assessing how suitable they are to receive crops based on the spatial constraints. In the case of the basic configuration:

$$S = \frac{40}{|40+40+20|} \times KD + \frac{40}{|40+40+20|} \times NA + \frac{20}{|40+40+20|} \times SW \quad (3)$$

where the relative contribution of kernel density (KD, 40), nutrient availability (NA, 40) and soil workability (SW, 20) are specified in Table 6. Assuming that the more suitable a grid-cell the more intensification it will receive, the 70km$^2$ of potential conversion are tentatively distributed to the selected grid-cells according to their suitability index:

$$TC_{gc} = 70km^2 \times \frac{S_{gc}}{\sum_{gcs} S} \quad (4)$$

where $TC$ stands for tentative conversion, $gc$ the grid-cell and $gcs$ the group of grid-cells selected because they contain both crops and grass. The actual conversion is the minimum value between $TC$ and the area of grass on each grid-cell:

$$AC = min(TC|GA) \quad (5)$$

where $AC$ stands for actual conversion and $GA$ for grass area. Because the tentative conversion might not be possible because the existing grass area is lower, the total conversion achieved after the first computation can be less than 70km$^2$ while some grid-cells might still have grass to spare. The computation is therefore repeated until 70km2 is reached, or until there are no more grid-cells with pre-existing crops and with grass to spare. For this example, we will assume that 60km$^2$ of grass was converted to crops. The total intensification target being 80km$^2$, 20km$^2$ has to be done on non-grass areas. The code thus goes to the next crop transition priority (Table 5): shrublands is skipped as it remains unchanged, but forests are projected to decrease by 30km$^2$. The intensification computation is repeated to convert forests to crops. We will assume that only 7km$^2$ of forest-to-crop conversion could be achieved. At the end of the intensification, 67km$^2$ of the projected 100km$^2$ cropland increase has been allocated, 60km$^2$ by converting grasslands, 7km$^2$ by converting forests, leaving 33km$^2$ to be done by expansion. Of those 33km$^2$, 10km$^2$ have to replace grasslands as they are projected to decrease by 70km$^2$ total, 60 of which have been converted by crop intensification. The remaining 23km$^2$ have to replace forests as they are projected to decrease by 30km$^2$, 7 of which have been converted by crop intensification.



The expansion function is similar to the intensification function, except that only a fraction of the pre-selected grid-cells are selected to receive the remaining crop area, to avoid patterns of ubiquitous expansion, especially for croplands which generally expand in clusters around newly cultivated lands. Continuing with the applied example, the code goes through the same treatment order and transition priorities, thus first expanding crops into grasslands. Because crop expansion can only occur on grid-cells that do not yet have crops, all grid-cells in the considered region/AEZ containing grass but not crops are pre-selected as "candidate" grid-cells to apply the grass conversion to crops. The candidate grid-cells are then ranked based on their suitability index. The code selects the most suitable candidate grid-cells following a percentage defined by the user (selection threshold, 25% in the basic configuration) or - if stochastic selection is turned on - using the suitability index of each grid-cell as the probability it will be selected in a Bernoulli trial. Only the selected grid-cells will be used when computing the tentative and actual conversion, in the same way as for intensification. 10km$^2$ of crops are expanded following this scheme into grasslands, and 23km$^2$ into forests.

The final intensification ratio is thus $67/100 = 0.67$, less than the user-defined target (0.8 in the basic configuration) because of a land-driven limitation on the amount of intensification that can be achieved. In other cases, the reverse situation occurs and a larger proportion of LULC change has to be done through intensification, conducted in the second round of intensification (see Figure 3).

At the end of a full downscaling run, gridded LULC areas are obtained for each of the user-defined land types and for each time step, which can be interpolated to annual data. LULC transitions (the amount of land switched between each pair of land types in each grid-cell) are also explicitly tracked by the algorithm given their importance for the carbon cycle (Brovkin et al., 2013). They are however not exported by default given their large size and specific format requirements as input for models capable of using them.

**2.3 Downscaling Evaluation and sensitivity analysis on historical land use change**

The downscaling code was applied to historical LULC change data with a range of alternative configurations to evaluate how realistic the spatial allocation is and to quantify the sensitivity of the results to the user-defined parameters.

Gridded estimates of historical land use from the HYDE database (version 3.1) were combined to gridded estimates of potential vegetation from the SAGE database to create base-year maps of LULC as inputs to the downscaling code. Although the HYDE data include estimates back to 10000 B.C., we contain the evaluation analysis to the 1700-2005 period. About 81% of today's cropland area in the database has been established within that period. Note that HYDE is a reconstruction product and shows substantial deviations from satellite data such as MODIS (Sect. 2.2). It is used as what we considered the best option to evaluate the downscaling, especially due to the temporal span. Using a satellite product directly





(e.g. MODIS) would restrain the evaluation to a 10-20 years period with much less land use change. The ratio of misclassified over real land use change would be an issue, with for example up to an order of magnitude between apparent versus real change in the MODIS annual land cover product (Friedl et al., 2010).

Six alternative configurations were defined for the 10 parameters shown in Table 7 (see also Sect. A in SM). The downscaling was thus run 60 times, each time maintaining the basic configuration but changing the value of the considered parameter.

The 2005 downscaled crop distribution was then compared to the HYDE data to compute a performance metric:

$$metric = 1 - \frac{\sum_{gc}|GCAMdc_{gc,2005} - HYDEc_{gc,2005}|}{\sum_{gc}|HYDEc_{gc,2005} - HYDEc_{gc,byear}|} \tag{6}$$

where *GCAMdc* is the downscaled GCAM crop area, *HYDEc* the crop area in the HYDE data, and *byear* is the base-year of the run (first time step). The metric is computed for single or aggregated regions/AEZs and can vary from -1 to +1. It represents the fraction of land use change that was allocated to the "right" grid-cells, i.e. in agreement with the land use change computed from HYDE between the base year and 2005. At the extremes, -1 means the downscaling did exactly the opposite to what was observed (decreasing crops where it was supposed to increase and vice-versa). A value of +1 is a

perfect match. The metric was computed for tropical, temperate and boreal climate zones separately, as delineated in Figure 1b based on the AEZs, as well as for each of the 32 regions delineated in Figure 1a.

### 2.4 Configuration for future projections

We applied the downscaling - with its basic configuration - to two GCAM scenarios of future LULC projections to illustrate the capabilities of the algorithm. In the "reference" scenario, the economy and technological developments are not targeted

by any environmental policies, thus driving human- and natural-system dynamics in a business-as-usual fashion. This scenario features substantial crop expansion to meet the increasing food demand (growing population, diet changes, etc). In the Mitigation Policy 4.5 (MP4.5) scenario, an economic policy in the form of a global carbon market applied to industrial, fossil-fuel and terrestrial emissions is implemented to limit radiative forcing in 2100 to +4.5W.m$^{-2}$. Incentives to carbon sequestration in the MP4.5 lead to afforestation in many regions, while agriculture tends to develop in high-yield areas. It is

a replicate of the Representative Concentration Pathways 4.5 scenario (RCP4.5) produced by GCAM for the IPCC 5[th] assessment report (Thomson et al., 2011, p.4), but instead uses the latest GCAM release (GCAM4).



### 3. Results

#### 3.1 Evaluation and sensitivity

The historical downscaling of LULC change starting from the 1800 base-year is presented in Figure 4 as one case illustrating the evaluation and sensitivity analysis. In 1800, parts of India, China and Europe already had substantial agriculture, but

North and South America, Africa, and Australia were still largely uncultivated (Figure 4a). For the regions where the main patterns of current agriculture were already defined, the downscaling reproduces agricultural trends quite well, including the crop intensification in India, China, Europe and Russia, albeit with a tendency for more spatial aggregation (Figure 4b,c). In North America, although the algorithm has little information on where agriculture will develop from the base-year, the Corn Belt and surrounding agricultural areas are somewhat reproduced, again with more aggregation. Similar results are obtained

in Australia, while the agricultural patterns are quite different from observations in South America and Africa. The performance metric computed at regional scale substantiates the visual finding of Figure 4: the algorithm performs better in regions where agricultural activities were well established in 1800 (e.g. India and Europe, Figure S2).

The downscaling performance is primarily sensitive to the historic base-year and to the aggregation of output to coarser

resolutions (Figure 5). As expected, the more recent the base-year, the better the performance: the algorithm has a better handle on the 2005 crop distribution if it starts with a 1980 map rather than a 1700 map. This is a consistent finding, either at the biome scale (Figure 5), or at the regional scale (Figure S2 in SM). The sensitivity to the resolution of the performance assessment is another expected finding: land use allocation attributed to the wrong 0.25 degree grid-cells are more likely to be aggregated to the right grid-cell at coarser resolution. Among the parameters governing the downscaling itself, the

performance is most sensitive to the intensification ratio and to the grid-cell selection process for land use expansion (selection threshold). Intermediate sensitivity is inferred for spatial drivers (kernel density, nutrient availability and soil workability), while the chosen sequence of land categories (treatment order and transition priorities) has little impact on the outputs.

#### 3.2 Future land use change scenarios

The downscaled reference and MP4.5 LULC scenarios feature key differences in 2100 due to their specific economic and policy context (Figure 6). Population reaches about 9 billion in 2060 in GCAM (slowly decreasing thereafter), contributing to increasing food demand that cannot be met with the projected yield improvements on current agricultural areas. As a result, global crop area is projected to increase by 10% in the reference scenario (2005-2100), with substantial expansion in tropical and sub-tropical forests (Figure 6c,d), compensated by afforestation in other regions (0.3% deforestation globally).

In the MP4.5 scenario, economic incentives for terrestrial carbon sequestration lead to a different solution. Afforestation becomes a profitable option for landowners and global forested area increases by 34%, replacing agriculture in many regions



(Figure 6e,f). To meet global food demand, agricultural production is intensified in high-yield areas (e.g. India, China) and expands into marginal lands with the support of irrigation and other technological developments (e.g. Western US, Middle East). Globally, those changes of agricultural practices enable a reduction of crop area by 10.4%. Note that these general LULC trends are determined by GCAM, including deforestation and afforestation: the downscaling does not make any

5  region/AEZ-scale land use change decision, but instead spatially delineates those decisions to a gridded format. Note also that the downscaled 2005 GCAM crops show much more similar patterns to MODIS than those obtained in the historical evaluation analysis (Figure 6a versus Figure 4). This is because 2005 was the last year in the evaluation analysis while it is the base-year for the projections.

## 4. Discussion

GCAM models human and natural systems at the scale of regions and AEZs, but the LULC downscaling system presented here enables a gridded representation of the land. The gridded outputs are consistent with the GCAM projections and can be influenced with a number of user-defined parameters. The algorithm relies on base-year observation-derived LULC distribution and performs best when key spatial features are already established in the base-year (e.g. agricultural areas). The

optional spatial constrains however provide the capability to adjust the downscaling to capture regional land-use change dynamics. One example illustrated in this study consists in using soil characteristics (i.e., nutrients, workability) to drive the allocation of agricultural activities. The downscaling system was primarily developed as an integrating tool, enabling LULC change impact assessments by providing gridded inputs to other models that cannot be run with the original GCAM data at region/AEZ scale. The gridded outputs may also be used to directly analyze spatial patterns within and between GCAM

scenarios, albeit with the understanding that realistic results depend on the chosen configuration for the region, time period and aspects of LULC that are being considered.

Models that might be coupled to GCAM through LULC downscaling all have specific input data requirements - different land types or spatial resolution for example. The downscaling system can be easily adapted to meet a number of these

requirements, without any edits to the code. Any number of final land types (FLTs) can be downscaled provided that base-year data is available and that parameters specific to each land type are provided (e.g. aggregation rules, transition priorities). For example, in the case of a model requiring separate broadleaf/needle-leaf and deciduous/evergreen forest types – for which base-year distribution is readily available in MODIS - Tables 2-6 need to be modified into Tables S2-S6 (supp. Material) and all 4 types of forests will be downscaled. Another example consists in downscaling specific crop types for

agricultural models (instead of a single crop category as shown in this study), which requires crop-specific base-year data and modified Tables 2-6. The necessary data and configuration tables are provided as a beta-version with the downscaling




system (see user manual). Other aspects that are flexible include the downscaling resolution, the spatial domain (e.g. continental, regional focus), and the base-year LULC data (e.g. higher quality regional datasets).

The downscaling method can be used for a number of applications related to LULC change. It contributed to a study
assessing global gridded carbon fluxes from agricultural production and consumption (Wolf et al., 2015), by downscaling FAO crop inventory data at the county/state/province level to a 0.05 degree grid. It was also applied to downscale specific crop types as well as irrigation practices from a detailed USA version of GCAM (Huang et al., in prep) to study the response of terrestrial hydrology to future scenarios of LULC change.

Despite the flexibility, some aspects are intrinsic to the GCAM model and the downscaling code and might be a limitation for certain applications. GCAM models net LULC change within each spatial unit (e.g. region/AEZ), as opposed to gross LULC, and thus minimizes the amount of LULC change from one timestep to the other. For example, a crop increase by $100km^2$ in a region/AEZ could be the result of several LULC change storylines, including one where crops increase by $150km^2$ in some part of the region/AEZ and decreased by $50km^2$ elsewhere. In that case $200km^2$ of land are converted (gross
change), with consequences for the carbon emissions, landscape fragmentation or the water cycle. The downscaling does not try to model those dynamics a posteriori from the GCAM net land use change. A new version of GCAM is currently being developed to represent land use dynamics annually, which the downscaling algorithm can process. The net versus gross LULC change issue will be mostly eliminated for these scenarios as multiple land conversions within the same year are rare. The lack of flexibility to account for specific dynamics through the regional and temporal domain of the downscaling is
another limitation, which can be addressed with code changes. As shown in the evaluation and sensitivity analysis, regional performances vary depending on the amount and type of observed LULC change and on the period considered. For example, patterns of agricultural expansion into the Amazon forest will be best downscaled under a specific configuration that would be sub-optimal to represent intensification and encroachment into semi-arid areas in the U.S. Great Plains. Similarly, that same configuration would not be the best to reproduce a move towards intensification and away from deforestation observed
in the Amazon basin since 2004. The parameterization is currently common to all regions and for the entire downscaling period, but can be made flexible with relatively simple edits to the code.

### 5. Data availability

The downscaling system is available, at https://github.com/JGCRI/GCAMLU, including a detailed user manual.



## 6. Author contribution

YLP and TW designed the model. YLP developed the model and performed the simulations. YLP prepared the manuscript with contributions from all authors.

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





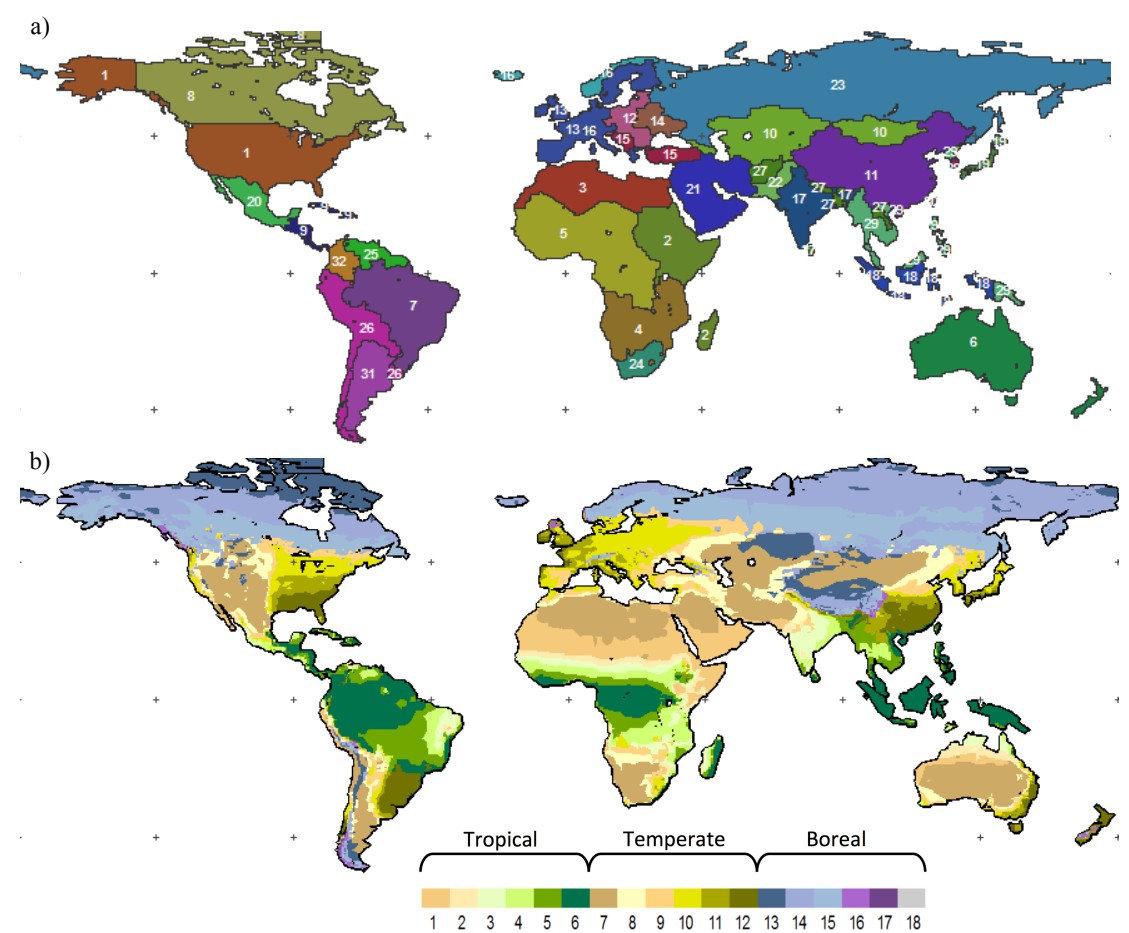

Regions name for figure a)

| | | | |
|---|---|---|---|
| 1- USA | 9- Central America and Caribbean | 17- India | 25- South America_Northern |
| 2- Africa_Eastern | 10- Central Asia | 18- Indonesia | 26- South America_Southern |
| 3- Africa_Northern | 11- China | 19- Japan | 27- South Asia |
| 4- Africa_Southern | 12- EU-12 | 20- Mexico | 28- South Korea |
| 5- Africa_Western | 13- EU-15 | 21- Middle East | 29- Southeast Asia |
| 6- Australia_NZ | 14- Europe_Eastern | 22- Pakistan | 30- Taiwan |
| 7- Brazil | 15- Europe_Non_EU | 23- Russia | 31- Argentina |
| 8- Canada | 16- European Free Trade Association | 24- South Africa | 32- Colombia |

**Figure 1: . a) 32 GCAM regions. b) Agro-Ecological Zones (AEZs).**





**Table 1. Terrestrial land types in GCAM.**

| | |
|---|---|
| **Natural land** | Forests, Shrublands, Grasslands, Tundra, Deserts |
| **Managed land** | Corn, Wheat, Rice, Root & Tubers, Oil crops, Sugar crops, Other grain crops, Fiber crops, Fodder grass, Fodder herb, Biomass crops, Miscellaneous crops, Other Arable Land (e.g. fallow), Palm fruit, Pasture, Urban, Willow, Managed forest. |



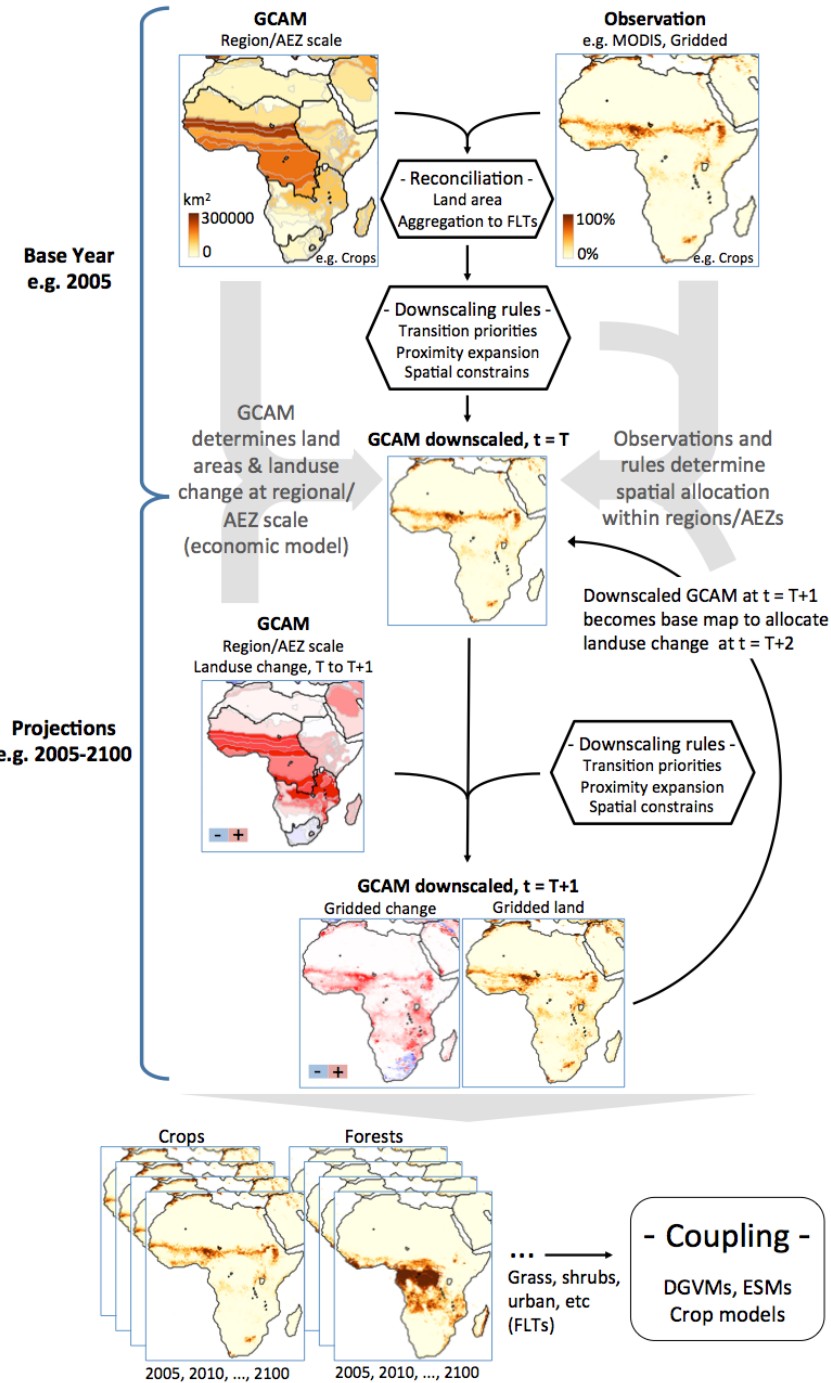

**Figure 2: . a) 32 GCAM regions. b) Agro-Ecological Zones (AEZs).**



**Table 2. User-defined aggregation of the spatial data.**

| Spatial land types (MODIS PFTs) | Common land types for downscaling | | | | | | |
|---|---|---|---|---|---|---|---|
| | forest | shrub | grass | crops | urban | snow | sparse |
| ev. needl. forest | **1** | 0 | 0 | 0 | 0 | 0 | 0 |
| ev. broad. forest | **1** | 0 | 0 | 0 | 0 | 0 | 0 |
| dec. needl. forest | **1** | 0 | 0 | 0 | 0 | 0 | 0 |
| dec. broad. forest | **1** | 0 | 0 | 0 | 0 | 0 | 0 |
| shrub | 0 | **1** | 0 | 0 | 0 | 0 | 0 |
| grass | 0 | 0 | **1** | 0 | 0 | 0 | 0 |
| crops | 0 | 0 | 0 | **1** | 0 | 0 | 0 |
| urban | 0 | 0 | 0 | 0 | **1** | 0 | 0 |
| snow | 0 | 0 | 0 | 0 | 0 | **1** | 0 |
| sparse | 0 | 0 | 0 | 0 | 0 | 0 | **1** |





**Table 3. User-defined aggregation of the GCAM data.**

| GCAM land types | Common land types for downscaling | | | | | | |
|---|---|---|---|---|---|---|---|
| | forest | shrub | grass | crops | urban | snow | sparse |
| Corn | 0 | 0 | 0 | **1** | 0 | 0 | 0 |
| Wheat | 0 | 0 | 0 | **1** | 0 | 0 | 0 |
| Rice | 0 | 0 | 0 | **1** | 0 | 0 | 0 |
| Root_Tuber | 0 | 0 | 0 | **1** | 0 | 0 | 0 |
| OilCrop | 0 | 0 | 0 | **1** | 0 | 0 | 0 |
| SugarCrop | 0 | 0 | 0 | **1** | 0 | 0 | 0 |
| OtherGrain | 0 | 0 | 0 | **1** | 0 | 0 | 0 |
| FiberCrop | 0 | 0 | 0 | **1** | 0 | 0 | 0 |
| FodderGrass | 0 | 0 | 0 | **1** | 0 | 0 | 0 |
| FodderHerb | 0 | 0 | 0 | **1** | 0 | 0 | 0 |
| Biomass | 0 | 0 | 0 | **1** | 0 | 0 | 0 |
| MiscCrop | 0 | 0 | 0 | **1** | 0 | 0 | 0 |
| OtherArableLand | 0 | 0 | 0 | **1** | 0 | 0 | 0 |
| PalmFruit | 0 | 0 | 0 | **1** | 0 | 0 | 0 |
| Pasture | 0 | 0 | **1** | 0 | 0 | 0 | 0 |
| UnmanagedPasture | 0 | 0 | **1** | 0 | 0 | 0 | 0 |
| UrbanLand | 0 | 0 | 0 | 0 | **1** | 0 | 0 |
| Willow | **1** | 0 | 0 | 0 | 0 | 0 | 0 |
| Forest | **1** | 0 | 0 | 0 | 0 | 0 | 0 |
| UnmanagedForest | **1** | 0 | 0 | 0 | 0 | 0 | 0 |
| Shrubland | 0 | **1** | 0 | 0 | 0 | 0 | 0 |
| Grassland | 0 | 0 | **1** | 0 | 0 | 0 | 0 |
| Tundra | 0 | 0 | **1** | 0 | 0 | 0 | 0 |
| RockIceDesert | 0 | 0 | 0 | 0 | 0 | **1** | **1** |



**Table 4. User-defined treatment order.**

| Common land types | Treatment order |
|---|---|
| forest | 5 |
| shrub | 7 |
| grass | 6 |
| crops | 4 |
| urban | 1 |
| snow | 2 |
| sparse | 3 |

5   **Table 5. User-defined transition priorities.**

| Common land types | Common land types for downscaling | | | | | | |
|---|---|---|---|---|---|---|---|
| | forest | shrub | grass | crops | urban | snow | sparse |
| forest | 0 | 1 | 2 | 3 | 4 | 6 | 5 |
| shrub | 5 | 0 | 4 | 6 | 1 | 3 | 2 |
| grass | 6 | 5 | 0 | 4 | 1 | 3 | 2 |
| crops | 4 | 3 | 2 | 0 | 1 | 6 | 5 |
| urban | 5 | 3 | 2 | 4 | 0 | 6 | 1 |
| snow | 6 | 5 | 4 | 3 | 2 | 0 | 1 |
| sparse | 6 | 4 | 2 | 3 | 5 | 1 | 0 |

**Table 6. User-defined spatial constrains.**

| Spatial constraints | Common land types for downscaling | | | | | | |
|---|---|---|---|---|---|---|---|
| | forest | shrub | grass | crops | urban | snow | sparse |
| Kernel density | 100 | 100 | 100 | 40 | 100 | 100 | 100 |
| Nutrient availability | 0 | 0 | 0 | 40 | 0 | 0 | 0 |
| Soil workability | 0 | 0 | 0 | 20 | 0 | 0 | 0 |



**1. Initialization**

# Reading in data/parameters
# Reconciliation (Land area in GCAM and MODIS)
# Aggregation to final land types (FLTs)

**2. Main code**

. **For each timestep:**
  . **For each land type:**
    # Compute kernel density
  . **For each Region:**
    . **For each AEZ:**
      . **For each land type $FLT_1$ (treatment order):**
        . **If $FLT_1$ increases:**
          . **For each one of the other land types $FLT_2$ (transition priorities):**
            . **If $FLT_2$ decreases:**
              # **Intensification:** increase $lt_1$ area on grid-cells where $lt_1$ and $lt_2$ exist, converting $FLT_2$ to $FLT_1$, based on spatial constrains.
        . **When intensification ratio or no more intensification possible:**
          # Break and go to next land type in treatment order

  . **For each Region:**
    . **For each AEZ:**
      . **For each land type $FLT_1$ (treatment order):**
        . **If $FLT_1$ increases:**
          . **For each one of the other land types $FLT_2$ (transition priorities):**
            . **If $lt_2$ decreases:**
              # **Expansion:** increase $FLT_1$ area on grid-cells where $FLT_1$ does not exist and $lt_2$ exist, converting $FLT_2$ to $FLT_1$, based on spatial constrains.
        . **When $FLT_1$ target achieved or no more expansion possible:**
          # Break and go to next land type in treatment order

  . **For each Region:**
    . **For each AEZ:**
      . **For each land type $FLT_1$ (treatment order):**
        . **If $lt_1$ target has not been achieved yet:**
          . **For each one of the other land types $FLT_2$ (transition priorities):**
            . **If $FLT_2$ decreases:**
              # **Second round of intensification:** increase $FLT_1$ area on grid-cells where $FLT_1$ and $FLT_2$ exist, converting $FLT_2$ to $FLT_1$, based on spatial constrains. After this round, all regions/AEZs/land-types targets are achieved.

**Figure 3. Save downscaled GCAM landuse and landcover**



**Table 7. Parameters and groups of parameters selected for the sensitivity analysis. *stochastic (see text). For each parameter, the model is run 6 times with alternative values for that parameter (a to f), while maintaining the default**
5  **parameterization for everything else.**

| Parameter | Default value | Alternative parameterizations | | | | | |
|---|---|---|---|---|---|---|---|
| | | a | b | c | d | e | f |
| Intensification ratio | 0.8 | 1 | 0.8 | 0.6 | 0.4 | 0.2 | 0 |
| Treatment order | Table 4 | Table S1 in Supp. Mat. | | | | | |
| Transition priorities | Table 5 | Table S2 in Supp. Mat. | | | | | |
| Selection threshold | 25% | 5 | 10 | 25 | 50 | 75 | stoch* |
| Kernel density radius | 10 | 2 | 4 | 10 | 25 | 50 | 100 |
| Kernel density constrain weight | 40 | 0 | 20 | 40 | 60 | 80 | 100 |
| Nutrient availability constrain weight | 40 | 0 | 20 | 40 | 60 | 80 | 100 |
| Soil workability constrain weight | 20 | 0 | 20 | 40 | 60 | 80 | 100 |
| Base year | 1800 | 1700 | 1800 | 1900 | 1950 | 1980 | 2000 |
| Evaluation resolution | 0.25 | 0.25 | 0.5 | 0.75 | 1 | 2 | 4 |





**Figure 4. Cropland distribution obtained when downscaling region/AEZ scale land use change from 1800 to 2005 under the basic configuration. Note that HYDE is used as the best option for a historic evaluation, but shows some discrepancies in 2005 with satellite-derived products such as the MODIS land use and land cover.**





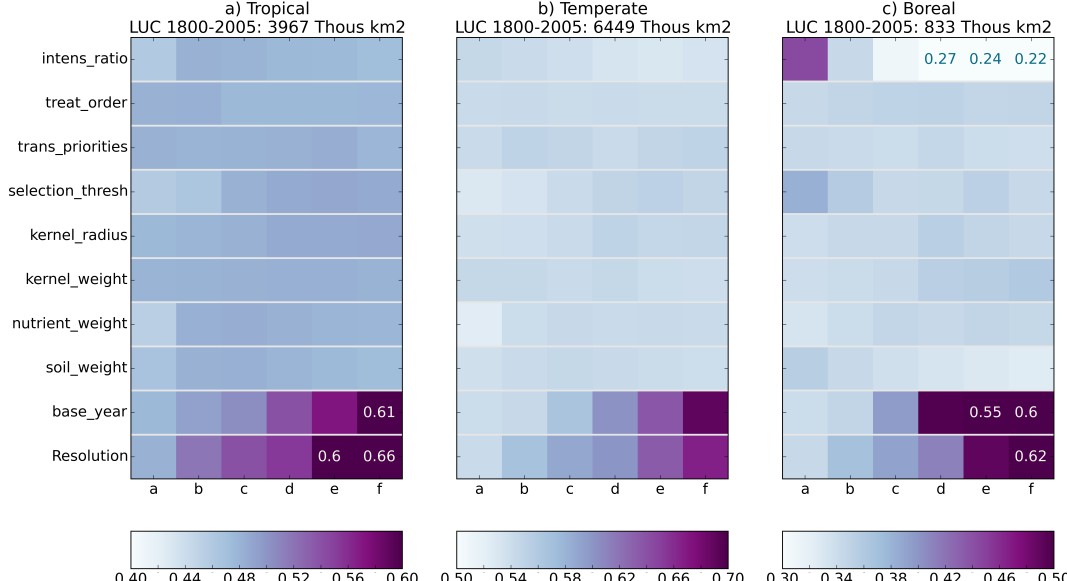

**Figure 5. Results of the sensitivity analysis. For the 6 alternative values (x-axis) of each parameter (y-axis), the color scale indicates the land use downscaling performance metric averaged over all grid-cells in the a) tropical, b) Temperate and c) Boreal biomes. Colorbar ranges are different for each biome. Beyond-range values are printed in the corresponding x,y position (the full scale of the metric is [-1 1], see text). Note: the figure uses a "perceptual" colormap with monotically decreasing lightness throughout the colormap. Equal changes in the evaluation metric anywhere are perceived as equal steps in the color space. Results of the sensitivity analysis per regions (32 GCAM regions) are provided in supplementary material (figure number).**





Figure 6. Downscaled crop and forest change in the GCAM reference and MP4.5 scenarios. Note that these scenarios are downscaled with the MODIS land cover product as base-year land use patterns, hence the differences between the 2005 map in a) and the 2005 map in Figure 4 which is obtained from the downscaling of historical land use change based on the HYDE product.