# Peer review of "Downscaling land use and land cover from the Global Change Assessment Model for coupling with Earth system models."

_Geoscientific Model Development, 2016_

## Referee Comment (RC1) · Anonymous Referee #1 · 5 Jul 2016

[General comments] Authors present in this paper a set of algorithm to spatially down-scale global land use change dataset that is simulated by GCAM, a regional/AEZ-based integrated assessment model, into gridded formats that are more familiar with Earth system / land ecosystem modelers. The basic idea and overview of the down-scaling algorithm are firstly introduced, and then the detailed procedure in the system is explained in source-code level. They execute sensitivity tests of the downscaling system, by applying the system to a historical land use change. Demonstrations of downscaling for future LULC (land use and land cover) scenarios are also introduced, with discussions on the potential applications and limitations of their systems. The algorithm and system introduced here are clearly important, because land use change is

one of the key issues that make linkages between scenario making, climate projection with the Earth system models, and impact assessments by land/agricultural models. The system introduced in this paper will help to bridge the research works between them. Thanks to the authors' careful descriptions on the downscaling procedure, this paper will help to understand the creation of LULC datasets simulated by GCAM. The system is well designed for general usages of downscaling and being available for everyone. No logical fault is found in this paper, but I think there are rooms to be improved, and they are listed below. Most of them will not require so much effort to improve.

[Detailed comments] p3, L17 "energy demand (biomass crops)" Is "energy demand (bioenergy crops)" more adequate?

P4, L1 "gridded LULC data" It seems better to note that this data is observation-based, not simulated by GCAM.

Fig.2 Figure title is as same as Fig.1, and thus should be changed. Because the figure outlines all downscaling algorithms, I hope enough explanation to be put in the caption. Specifically, readers will read the manuscript more easily if you can put in the caption the linkages between technical words ("reconciliation", "transition priorities", "proximity expansion", etc) and the subsection number: e.g. "land area matching in the reconciliation process is shown in 2.2.2.1".

P5 L8 The subsection number "2.2.2.1" should be replaced by "2.2.2.2".

P5 L10 "cropland PFT" should be replaced by "cropland plant functional type (PFT)", or simply "cropland type". In my thinking, since you have already used three types of categories ("GLTs", "SLTs", and "FLTs") for land types, additional use of "PFT" will make readers confused.

Table 2 and 3: "Final land types (FLTs) for downscaling" looks better for the column title, and it will be helpful for readers if there are brief explanations in the caption on how to read this table.

[Figure]

P5 L29-30 In Fig.2, "transition priorities" and "spatial constraints" are shown, but "treatment order" and "intensification versus expansion ratio" likely not. It will be helpful for readers if you can put the latter two items on the figure.

P6, L7: "Intensification versus expansion ratio" looks better for the title, as you describe in p5, L31.

P6 subsection 2.2.3.4: In my view, the definition of index S ("suitability index"), which first appears in 2.2.4, should be done in this subsection, because readers cannot imagine how the KD, NA, and SW work to constrain the spatial distribution. In addition, NA and SW in eq (3) seem to have units with dimensionless: please specify them in the text.

P8, L2: "see Sect. 1.2.3.2" should be changed to "see Sect. 2.2.3.2".

Fig.3: 1) I found no loop for global grid cells. Does the loop represented by "For each land type" correspond to the loop? 2) I can find the terms "lt1" and "lt2" in the figure, but there is no explanation for them. Do they respectively represent "FLT1" and "FLT2"? 3) The figure title is ambiguous for me, and should be replaced by adequate one?

P9, L23- I'm not sure about the target dataset to which the downscaling method was applied. Did you use HYDE information that was aggregated into regional/AEZ map, and then apply the downscaling algorithm to it? Or have you used historical LULC changes simulated by GCAM? In the latter case, it might be better to refer to the existing work that created the LULC changes.

P10, L18- About initial condition for the projection: Did you use HYDE for base-year-map in 2005? Or MODIS?

P11, section 3.1 I hope to see some description on the score of the metric in the basic configuration run. Although we can see in Fig. 4 the spatial distribution of the result with the configuration, we are not sure how the basic configuration did reasonable job in the metric.

[Figure]

About figure: it seems slight curious for me that we cannot see any comparison of maps between "before downscaling" and "after-downscaling" throughout this paper, although downscaling is the main topic. In my simple thinking, such maps would attract attention from the readers who are not so familiar with integrated assessment models or creation of LULC scenarios, and would visualize the significance of your downscaling work. I propose the authors to put such maps in supplementary materials (or main body).

---

## Referee Comment (RC2) · Anonymous Referee #2 · 12 Jul 2016

[General comment]

This paper describes the downscaling algorithm to generate the gridded data from the regional data calculated by the Global Change Assessment Model. After the explanation of the algorithm, methods of the downscaling evaluation and the sensitivity analysis is described, and finally, the result is evaluated.

Downscaling technique is one of a main topic of climate simulation, this paper will be helpful for understanding the concepts and ideas of this technique. Description of the downscaling algorithm and evaluation methods is so detailed and polite that readers can easily understand it.

[Figure]

But the analysis results seem to be insufficient to show usefulness and advantage of this model. In addition, some more detailed descriptions and modifications seem to be required for better understanding.

[Major comments]

<The result of parameter sensitivity test>

Evaluation of parameter sensitivity summarized in Fig.5 is main topic of this paper. The result, as the author said, is dominated by mainly base year and grid resolution, and sensitivities of other parameters are relatively low. The problem is that, under the default value of base year (1800) and resolution (0.25), the result in Fig.5 can be interpreted that the reproductivity is not so good and this poor reproductivity cannot be improved by changing of any other parameters.

Therefore, I strongly recommend to recompute the parameter sensitivity under the practical base year (1900 or 1950) and resolution (0.25) setting, and redraw the Fig.5 in appropriate color scale without base_year and resolution to show the sensitivity of other parameters clearly.

The author shows only the result of crop, but it seems to be insufficient to insist that the downscaling algorithm is really useful. I think that it is necessary to show the result of other land use, at least, the forest case that strongly affects on carbon cycle.

Author mentioned about results of parameter sensitivities at P11, L19-L23, but this explanation seems to be too simple. A more detailed description is desired after the re-calculation and re-drawing.

<model description>

Model overview is described in section 2.1. But the description is totally insufficient. For example, the phrase "the terrestrial modules" (P2, L28) suddenly appears in the section title. Before this section, the author mentioned about "GCAM" and reader did not be given any information about module structure of GCAM. The phrase "Over the spin-up

period" (P3, L13) is also the same. The readers not familiar to GCAM cannot prefigure the existence of spin-up period. For better understanding of GCAM and downscaling system, at least, the whole structure of GCAM and the computational flow should be shown in some figures.

<configuration of chapters>

Both downscaling methods and evaluation method are described in section 2. But these methods are essentially different and both are respectively important, and, despite the importance, section number indent seems to be too deep.

Therefore, I think that it is better to separate the description of the downscaling method and evaluation method and summarize the evaluation method and results into new section. Also, model overview is important and is required more detailed description as mentioned above.

As a result, it is preferable to modify the structure of chapters as follows.

| before | after |
|---|---|
| 1 Introduction | 1 Introduction |
| 2 .1 Overview of the terrestrial | 2 Model Overview |
| 2.2 Downscaling method | 3 Downscaling algorithm |
| | 4 Evaluation and sensitivity analysis |
| 2.3 Downscaling evaluation and | 4.1 method |
| 3.1 Evaluation and sensitivity | 4.2 results |
| | 5 Future projections |
| 2.4 Configuration for future projection | 5.1 Objective and configuration |
| 3.2 Future land use change scenarios | 5.2 results |

Authors mentioned that the objective of future projection is to illustrate the capabilities of the algorithm (P10,L18). But this reason seems to be weak. If there is a little more detailed description, there might be more convincing.

\

This is a model description paper, so originality is not required so strongly. But generality of the problem and solution is also important for scientific and technical progress, and should not be ignored even in a model description paper.

The author mentioned that spatial resolution is a technical challenge (P2, L11), but only from this explanation, reader cannot judge how this challenge has generality on climate science. Therefore, I ask a presentation of previous studies and an explanation of more detailed background of this study.

[Minor comments]

P2,L19: Meaning of the brackets (Kraucunas et al., 2014) is not clear.

P2,L31: Correspondence of the brackets is wrong.

P3,L13: Does " (1700-2005)" have a specific meaning? If so , description is required. If not, it is an extra information.

P3,L29: land use and land cover -> LULC

P4,L15-L19: This paragraph should be moved to "Data availability" section.

P6,L16: The code can easily be modified.... If it is so easy, why do not you do so?

P8,L2: Sect. 1.2.3.2 -> Sect. 2.2.3.2.

P24,Table 7: This table summarizes the parameters about a key topic of this paper. So, it is desirable to show all information without omission. Authors should not expect that readers are so diligent as to refer to supplementary material while reading a paper.

---

## Author Comment (AC1) · 4 Aug 2016

**[General comments]** Authors present in this paper a set of algorithm to spatially downscale global land use change dataset that is simulated by GCAM, a regional/AEZ- based integrated assessment model, into gridded formats that are more familiar with Earth system / land ecosystem modelers. The basic idea and overview of the down- scaling algorithm are firstly introduced, and then the detailed procedure in the system is explained in source-code level. They execute sensitivity tests of the downscaling system, by applying the system to a historical land use change. Demonstrations of downscaling for future LULC (land use and land cover) scenarios are also introduced, with discussions on the potential applications and limitations of their systems. The algorithm and system introduced here are clearly important, because land use change is one of the key issues that make linkages between scenario making, climate projection with the Earth system models, and impact assessments by land/agricultural models. The system introduced in this paper will help to bridge the research works between them. Thanks to the authors' careful descriptions on the downscaling procedure, this paper will help to understand the creation of LULC datasets simulated by GCAM. The system is well designed for general usages of downscaling and being available for everyone. No logical fault is found in this paper, but I think there are rooms to be improved, and they are listed below. Most of them will not require so much effort to improve.

We thank the reviewer for his helpful review, please find below our responses to comments and suggestions.

**[Detailed comments] p3, L17 "energy demand (biomass crops)" Is "energy demand (bioenergy crops)" more adequate?**
Yes indeed, we changed it.

**P4, L1 "gridded LULC data" It seems better to note that this data is observation-based, not simulated by GCAM.**
We now specify that the initial gridded data for the downscaling are observation-derived.

**Fig.2 Figure title is as same as Fig.1, and thus should be changed. Because the figure outlines all downscaling algorithms, I hope enough explanation to be put in the caption. Specifically, readers will read the manuscript more easily if you can put in the caption the linkages between technical words ("reconciliation", "transition priorities", "proximity expansion", etc) and the subsection number: e.g. "land area matching in the reconciliation process is shown in 2.2.2.1".**
We updated the caption accordingly:
"Overview of the downscaling method. The figure shows the successive computational steps to downscale a LULC change scenario from 2005 to 2100 described in the text (Sect. 2). The "Reconciliation" phase is detailed in Sect. 2.1 ; The "Downscaling rules" are detailed in Sect. 2.2, including the "treatment order" (Sect. 2.2.1), Intensification versus expansion ratio" (Sect. 2.2.2), the "Transition priorities" (Sect. 2.2.3) and the spatial constrains (Sect. 2.2.4)."

**P5 L8 The subsection number "2.2.2.1" should be replaced by "2.2.2.2".**
This was corrected while re-structuring the manuscript (see other reviewer's comments).

**P5 L10 "cropland PFT" should be replaced by "cropland plant functional type (PFT)", or simply "cropland type". In my thinking, since you have already used three types of categories ("GLTs", "SLTs", and "FLTs") for land types, additional use of "PFT" will make readers confused.**
Indeed, we replaced cropland PFT with "cropland type".

**Table 2 and 3: "Final land types (FLTs) for downscaling" looks better for the column title, and it will be helpful for readers if there are brief explanations in the caption on how to read this table.**
Changed

**P5 L29-30 In Fig.2, "transition priorities" and "spatial constraints" are shown, but "treatment order" and "intensification versus expansion ratio" likely not. It will be helpful for readers if you can put the latter two items on the figure.**
Thanks ! We added the missing downscaling rules in the figure.

**P6, L7: "Intensification versus expansion ratio" looks better for the title, as you describe in p5, L31.**
Changed

**P6 subsection 2.2.3.4: In my view, the definition of index S ("suitability index"), which first appears in 2.2.4, should be done in this subsection, because readers cannot imagine how the KD, NA, and SW work to constrain the spatial distribution. In addition, NA and SW in eq (3) seem to have units with dimensionless: please specify them in the text.**
We moved the definition of the suitability index to the "spatial constrain" section as suggested (note the overall change in structure following the other reviewer feedback). We also specify that S is dimensionless:
"Each spatial constrain being a dimensionless index bound from 0 to 1, the suitability index is dimensionless as well."

**P8, L2: "see Sect. 1.2.3.2" should be changed to "see Sect. 2.2.3.2".**
Corrected in the new structure.

**Fig.3: 1) I found no loop for global grid cells. Does the loop represented by "For each land type" correspond to the loop? 2) I can find the terms "lt1" and "lt2" in the figure, but there is no explanation for them. Do they respectively represent "FLT1" and "FLT2"? 3) The figure title is ambiguous for me, and should be replaced by adequate one?**
1) There is actually no loop on grid-cells. Once the algorithm reached a given Region/AEZ and FLT to be expanded, it then considers at once all potential grid-cells for the transition. Computations (e.g. suitability, area available for transition) are done with a 1-D array that contains those potential grid-cells.

2) The lt1 and lt2 were left from a previous draft of the figure, sorry about that, they are in fact FLT1 and FLT2.

3) The new title for the figure now reads: "Computation flow of the downscaling code."

**P9, L23- I'm not sure about the target dataset to which the downscaling method was applied. Did you use HYDE information that was aggregated into regional/AEZ map, and then apply the downscaling algorithm to it? Or have you used historical LULC changes simulated by GCAM? In the latter case, it might be better to refer to the existing work that created the LULC changes.**
Indeed that was not clear in the manuscript. We did the first case, we aggregated LULC change from HYDE data into Region/AEZ tabular data, from 1700 to 2005, and used these tabular data to run the downscaling. So GCAM was not part of the evaluation. It is now specified in the evaluation section of the manuscript:

"Gridded estimates of historical land use from the HYDE database (version 3.1) were combined to gridded estimates of potential vegetation from the SAGE database to create base-year gridded maps of LULC and Region/AEZ aggregated data of LULC change as inputs to the downscaling code."

**P10, L18- About initial condition for the projection: Did you use HYDE for base-year- map in 2005? Or MODIS?**
Again sorry that wasn't well specified, we used MODIS. It is now clearly stated in the text:
"Contrarily to the historical evaluation analysis that was using HYDE data for the base-year gridded LULC, the projection analysis starts in 2005 with observation-derived MODIS LULC. The downscaling is run with the default configuration presented in Sect 2."

**P11, section 3.1 I hope to see some description on the score of the metric in the basic configuration run. Although we can see in Fig. 4 the spatial distribution of the result with the configuration, we are not sure how the basic configuration did reasonable job in the metric.**
We now provide a little more detail on the performance metric. However, as mentioned in the text, HYDE is a reconstruction product and shows significant discrepancies with observation-derived LULC. The main application of the metric is for the sensitivity analyses.

"The historical downscaling of LULC change starting from the 1900 base-year is presented in Figure 5. Europe had already acquired most of today's cropland extent by 1900, but all other regions experienced a substantial increase in cropland area, both in the form of intensification (e.g. India) or expansion (e.g. North America). The downscaling algorithm leads to a spatial 2005 cropland distribution that is in general agreement with the HYDE data, yet lacking their smooth patterns (e.g. North America, India in Figure 5b,c). However, this smooth aspect seems to be an artifact of the HYDE data when compared to the MODIS data (Figure 5c and Figure 7a).

The performance metric generally ranges from 0.3 to 0.7 according to the region and configuration considered (Figure 6), indicating that the downscaling allocates fairly well the changes in cropland area (the metric is bound from -1 to 1). Performance and sensitivity to the downscaling parameters are quite different between tropical, temperate and boreal regions, indicating that LULC dynamics differ and cannot be captured by a single downscaling configuration. Overall, however, sensitivity to the intensification versus expansion ratio and to the relative contribution of kernel density are the strongest, suggesting the importance of proximity to pre-existing agricultural areas for the allocation of new crops. The performance of the downscaling is also clearly influenced

by the base-year, especially in the case of tropical regions, and, expectedly, by the aggregation of the output LULC to coarser resolution."

**About figure: it seems slight curious for me that we cannot see any comparison of maps between "before downscaling" and "after-downscaling" throughout this paper, although downscaling is the main topic. In my simple thinking, such maps would attract attention from the readers who are not so familiar with integrated assessment models or creation of LULC scenarios, and would visualize the significance of your downscaling work. I propose the authors to put such maps in supplementary materials (or main body).**

Indeed that was clearly missing, thanks ! We now added figure 2 below, showing the Regional/AEZ scale distribution of croplands, which can be compared to the downscaled maps in Figure 5 and Figure 7 as mentioned in caption.

[Figure]

**Figure 1. Distribution of 2005 GCAM croplands at the Region/AEZ scale. The algorithm presented in this paper downscales these patterns to a gridded scale (Figure 5 and Figure 7).**

---

## Author Comment (AC2) · 4 Aug 2016

**[General comment]**

**This paper describes the downscaling algorithm to generate the gridded data from the regional data calculated by the Global Change Assessment Model. After the explanation of the algorithm, methods of the downscaling evaluation and the sensitivity analysis is described, and finally, the result is evaluated.**

**Downscaling technique is one of a main topic of climate simulation, this paper will be helpful for understanding the concepts and ideas of this technique. Description of the downscaling algorithm and evaluation methods is so detailed and polite that readers can easily understand it.**

**But the analysis results seem to be insufficient to show usefulness and advantage of this model. In addition, some more detailed descriptions and modifications seem to be required for better understanding.**

We thank the reviewer for his helpful, thorough review of the manuscript. We discuss the suggested changes below and detail their implementation.

**[Major comments]**

**<The result of parameter sensitivity test>**

**Evaluation of parameter sensitivity summarized in Fig.5 is main topic of this paper. The result, as the author said, is dominated by mainly base year and grid resolution, and sensitivities of other parameters are relatively low. The problem is that, under the default value of base year (1800) and resolution (0.25), the result in Fig.5 can be interpreted that the reproductivity is not so good and this poor reproductivity cannot be improved by changing of any other parameters.**

**Therefore, I strongly recommend to recompute the parameter sensitivity under the practical base year (1900 or 1950) and resolution (0.25) setting, and redraw the Fig.5in appropriate color scale without base year and resolution to show the sensitivity of other parameters clearly.**

We now perform the sensitivity analysis with the 1900 base year, and made the sensitivity to most parameters more visible by reducing the colormap range. To avoid having two colorbars, we kept the sensitivity to base year and resolution in the same figure, with a text box indicating the metric value when it is out-of-range. The original script to generate the figure had an error, and the results for the sensitivity to base-year are now different. The discussion was updated as follow:

"The historical downscaling of LULC change starting from the 1900 base-year is presented in Figure 5. Europe had already acquired most of today's cropland extent by 1900, but all other regions experienced a substantial increase in cropland area, both in the form of intensification (e.g. India) or expansion (e.g. North America). The downscaling algorithm leads to a spatial 2005 cropland distribution that is in general agreement with the HYDE data, yet lacking their smooth patterns (e.g. North America, India in Figure 5b,c). However, this smooth aspect seems to be an artifact of the HYDE data when compared to the MODIS data (Figure 5c and Figure 7a).

The performance metric generally ranges from 0.3 to 0.7 according to the region and configuration considered (Figure 6), indicating that the downscaling allocates fairly well the changes in cropland area (the metric is bound from -1 to 1). Performance and

sensitivity to the downscaling parameters are quite different between tropical, temperate and boreal regions, indicating that LULC dynamics differ and cannot be captured by a single downscaling configuration. Overall, however, sensitivity to the intensification versus expansion ratio and to the relative contribution of kernel density are the strongest, suggesting the importance of proximity to pre-existing agricultural areas for the allocation of new crops. The performance of the downscaling is also clearly influenced by the base-year, especially in the case of tropical regions, and, expectedly, by the aggregation of the output LULC to coarser resolution."

**The author shows only the result of crop, but it seems to be insufficient to insist that the downscaling algorithm is really useful. I think that it is necessary to show the result of other land use, at least, the forest case that strongly affects on carbon cycle. Author mentioned about results of parameter sensitivities at P11, L19-L23, but this explanation seems to be too simple. A more detailed description is desired after there-calculation and re-drawing.**

The evaluation for forest change is now provided in supplementary material and discussed in the text:

"Performance was also evaluated for the downscaling of forests (Figure S2), which is a critical aspect for many environmental studies (e.g. carbon cycle, biodiversity). The results are mostly relevant for the tropical biome, where the evaluation shows similar patterns of sensitivity to those of croplands. Both temperate and boreal biomes experienced relatively little forest change from 1900 to 2005."

[Figure]

**Figure S2. Results of the performance and sensitivity analysis for forests.**

Note that we removed the 32 region sensitivity plots as they were not of much support to the results and discussion.

**<model description>**

**Model overview is described in section 2.1. But the description is totally insufficient. For example, the phrase "the terrestrial modules" (P2, L28) suddenly**

**appears in the section title. Before this section, the author mentioned about "GCAM" and reader did not be given any information about module structure of GCAM. The phrase "Over the spin-up period" (P3, L13) is also the same. The readers not familiar to GCAM cannot prefigure the existence of spin-up period. For better understanding of GCAM and downscaling system, at least, the whole structure of GCAM and the computational flow should be shown in some figures.**

An overview of the GCAM model is now part of the main manuscript, complementing the introduction paragraph and giving more details on the representation of the terrestrial biosphere. There is a figure showing the structure of GCAM and the overall computational flow (Figure S1) in supplementary materials, which we suggest to leave there given the more detailed description now part of the manuscript.

**<configuration of chapters>**

**Both downscaling methods and evaluation method are described in section 2. But these methods are essentially different and both are respectively important, and, despite the importance, section number indent seems to be too deep.**

**Therefore, I think that it is better to separate the description of the downscaling method and evaluation method and summarize the evaluation method and results into new section. Also, model overview is important and is required more detailed description as mentioned above.**

**As a result, it is preferable to modify the structure of chapters as follows.**

| before | after |
|---|---|
| 1 Introduction | 1 Introduction |
| 2 .1 Overview of the terrestrial | 2 Model Overview |
| 2.2 Downscaling method | 3 Downscaling algorithm |
| | 4 Evaluation and sensitivity analysis |
| 2.3 Downscaling evaluation and | 4.1 method |
| 3.1 Evaluation and sensitivity | 4.2 results |
| | 5 Future projections |
| 2.4 Configuration for future projection | 5.1 Objective and configuration |
| 3.2 Future land use change scenarios | 5.2 results |

Thanks! We updated the manuscript with the proposed structure.
* * *
**This is a model description paper, so originality is not required so strongly. But generality of the problem and solution is also important for scientific and technical progress, and should not be ignored even in a model description paper.**

**The author mentioned that spatial resolution is a technical challenge (P2, L11), but only from this explanation, reader cannot judge how this challenge has generality on climate science. Therefore, I ask a presentation of previous studies and an explanation of more detailed background of this study.**

The paper clearly lacked a paragraph introducing the importance of land use change projections for environmental studies, which we now address in the revised paper:

"LULC change is a key component of environmental change studies. More than 50% of the terrestrial biosphere has now been transformed to urban areas, croplands or rangelands by anthropogenic activities (Ellis, 2011). Estimates of the carbon budget from historical LULC change range from 12.5% to 33% of all anthropogenic carbon emissions depending on the time period and method considered (Houghton et al., 2012). These emissions combined to LULC-induced albedo and moisture dynamic alterations are a significant – albeit poorly constrained - climate forcing (e.g.Brovkin et al., 2013; Mahmood et al., 2010; Pongratz et al., 2010). The array of LULC changes impacts extends to many other environmental aspects, including biodiversity, freshwater resources and air quality (Foley et al., 2005), hence the importance of projecting future land use scenarios for impacts assessments."

Regarding the issue of spatial resolution, this is really a problem specific to GCAM, and can't be much more discussed than it is. We illustrate the sub-regional, non-regular spatial scale of the terrestrial component in GCAM (the combination of the 32 regions and 18 AEZs), present the approach developed to downscale GCAM LULC to a grid for the IPCC 5[th] assessment (Hurtt et al., 2011), and discuss the need for a GCAM-specific, flexible downscaling tool.

**[Minor comments]**

**P2,L19: Meaning of the brackets (Kraucunas et al., 2014) is not clear.**

Corrected

**P2,L31: Correspondence of the brackets is wrong.**

Corrected

**P3,L13: Does " (1700-2005)" have a specific meaning? If so , description is required. If not, it is an extra information.**

This was removed with the mention to the spin-up period (see former comment).

**P3,L29: land use and land cover -> LULC**

Done

**P4,L15-L19: This paragraph should be moved to "Data availability" section.**

Done

**P6,L16: The code can easily be modified.... If it is so easy, why do not you do so?**

We had to stop developing at some point, as there are many other additions that would be of interest for some users and applications (e.g. adding other constrains, having other regional or AEZ-specific parameterizations, etc). It wouldn't be an instant edit to the code though, so we removed "easily".

**P8,L2: Sect. 1.2.3.2 -> Sect. 2.2.3.2.**

Sorry about that, this was now changed according to the new structure of the manuscript.

**P24,Table 7: This table summarizes the parameters about a key topic of this paper. So, it is desirable to show all information without omission. Authors should not expect that readers are so diligent as to refer to supplementary material while reading a paper.**

The tables have been moved from supplementary to the main manuscript (Table 8 and 9).

---

## Author Response (AR2)

Dear Hisashi Sato,

Thank you for accepting the manuscript ! We have updated the caption of Tables 2 and 3.

Best regards,
Yannick Le Page and co-authors.